

# Analysis of selected variables in body composition, upper limb strength, and resting energy expenditure among youth soccer players: insights based on field position

Edyta Łuszczki[1], Anna Bartosiewicz[1], Katarzyna Dereń[1], Paweł Jagielski[2] and Andrzej Łukasik[1]

[1] Faculty of Health Sciences and Psychology, Medical College, University of Rzeszów, Rzeszów, Poland
[2] Department of Nutrition and Drug Research, Institute of Public Health, Faculty of Health Sciences, Jagiellonian University Medical College, Kraków, Poland

## ABSTRACT

**Background**. The study aimed to determine and compare selected variables of body composition, upper limb strength, and resting energy expenditure from the perspective of field position in youth soccer players.

**Methods**. The study group consisted of 189 boys aged 9 to 19 years ($\bar{x} = 14.7 \pm 2.6$ years) from the Sports Championship Schools. Body composition was determined using dual-energy X-ray absorptiometry (DXA), and resting energy expenditure (kcal/day) was measured using a Cosmed Quark RMR indirect calorimeter. Muscle strength was assessed using a handgrip dynamometer to measure grip strength.

**Results**. We studied 189 boys (mean body mass $58.4 \pm 16.1$ kg; mean height $166.9 \pm 14.5$ cm) divided by field position: attackers ($n = 43$), defenders ($n = 70$), goalkeepers ($n = 21$), and midfielders ($n = 55$). Significant between-position differences were observed in age, body mass, height, body mass index (BMI), grip strength (GS), lean mass, fat mass, bone mineral content (BMC), bone mineral density (BMD), and resting energy expenditure (REE). Body mass differed between attackers and goalkeepers (51.3 *vs.* 73.3 kg) and defenders and goalkeepers (58.7 *vs.* 73.3 kg). Goalkeepers had the highest body mass. REE measured by indirect calorimetry differed between attackers and goalkeepers (1,729 *vs.* 2,088 kcal/day).

**Conclusion**. These results allow for determining favorable parameters for each position on the field for young football players.

# INTRODUCTION

Soccer is one of the most widely played sports globally and holds particular significance in Poland, where it represents a substantial proportion of youth sports participation (*Central Statistical Office, 2012*). As the sport continues to evolve, both in physical demands and scientific interest, the need for targeted, evidence-based approaches to player development

Corresponding author
Edyta Łuszczki, eluszczki@ur.edu.pl

has increased. In particular, physical characteristics such as body composition, muscle strength, and energy expenditure have emerged as critical performance indicators, especially in youth athletes whose bodies are still maturing (*Bangsbo, 1994*; *Reilly, Bangsbo & Franks, 2000*; *Toselli et al., 2022*).

Each field position in soccer demands specific physiological and anthropometric traits (*Stølen et al., 2005*; *Slimani et al., 2018*; *Herbert et al., 2020*). Goalkeepers tend to be taller and heavier, often showing higher lean mass and energy requirements compared to midfielders or attackers (*Sutton et al., 2009*; *Sebastiá-Rico et al., 2023*). Identifying these position-specific profiles early in youth development is essential for optimizing training loads and reducing injury risks (*Collins et al., 2021*). Despite a growing body of research in adult and elite athletes, fewer studies focus on children and adolescents, especially using gold-standard assessment methods such as DXA and indirect calorimetry.

Body composition and resting energy expenditure (REE) are not only related to performance but also influence nutritional planning and recovery strategies (*Anderson et al., 2019*; *Clarke et al., 2011*). Grip strength, a proxy for general muscular fitness, has also gained attention in pediatric populations due to its predictive value for both athletic and general health outcomes (*Mehmet, Yang & Robinson, 2020*). However, there remains a lack of comprehensive studies combining these variables across different playing positions in young soccer players.

Therefore, the aim of this study was to determine and compare selected variables—body composition, upper limb strength, and resting energy expenditure—by field position in a cohort of male youth soccer players. By identifying position-specific physical characteristics at an early age, the findings can support talent identification, tailored training, and long-term athletic development strategies.

## MATERIALS & METHODS

### Participants
Data were collected as previously described in *Łuszczki et al. (2023)*. Specifically, participants were recruited in September 2021 from Sports Championship Schools in the Podkarpackie Voivodeship (south-eastern Poland). Thirteen schools were invited, and six agreed to take part. A total of 227 male students (aged 9–19 years) and their guardians provided consent; after excluding 38 boys who did not complete all assessments (due to injury, illness, or withdrawal), 189 participants remained for analysis. Eligible participants were male soccer players enrolled in a Sports Championship School with a minimum of 2 years of structured training (≥4 practices per week plus one official match weekly). Boys with any acute or chronic illness (including any febrile infection in the past 6 months), those taking medications or supplements affecting metabolism, or those who did not complete all measurements were excluded. All participants and their parents/guardians provided written informed consent before participation.

### Training description
All players followed a standardized training program as part of their Sports Championship School curriculum. Weekly training consisted of approximately 90 min of soccer-specific

field practice four times per week, ~45 min of gym-based motor skills training four times per week, and one 90-minute competitive match each week. All sessions were supervised by certified coaches, ensuring consistent training volume and intensity across participants.

## General procedures

All examinations and measurements were conducted in the university laboratory under controlled conditions, as previously described in *Łuszczki et al. (2023)*. Specifically, testing took place in the morning (between 7:00 and 10:00 a.m.) in a climate-controlled room maintained at 22–25 °C.

Participants were instructed to arrive in a rested, fasting state and received standardized instructions prior to assessments.

## Anthropometric measurements, body composition

Anthropometric assessments followed protocols previously described by *Sobek & Jagielski (2022)*. Specifically, standing height was measured without shoes using a portable stadiometer (Seca 213; accuracy 0.5 cm) with participants standing upright against the backboard. The mean of three consecutive height measurements was recorded for analysis. Body mass was measured to the nearest 0.1 kg using a calibrated digital scale, with participants in light clothing. Body composition (including total body bone mineral density, lean mass, and fat mass) was assessed *via* dual-energy X-ray absorptiometry (DXA) using a GE Healthcare Lunar iDXA scanner. Scans were performed by trained technicians according to the manufacturer's protocols, and results were analyzed using enCore software (v13.6). A calibration phantom was scanned periodically to ensure measurement accuracy. Low bone mineral density (BMD) was defined in accordance with *Łuszczki et al. (2021)*: a BMD $Z$-score between $-1.0$ and $-2.0$ was considered low BMD, and a $Z$-score $\leq -2.0$ at any age-matched reference site was considered very low BMD. BMI was calculated as weight (kg) divided by height squared ($m^2$). Weight status categories were assigned based on Polish age- and sex-specific BMI percentile norms: underweight (<5th percentile), normal weight (5th–85th percentile), overweight (85th–95th percentile), or obese ($\geq$95th percentile).

## Resting energy expenditure

Resting energy expenditure (REE) was measured by indirect calorimetry as previously described in *Łuszczki et al. (2023)*. Specifically, participants rested in a supine position while a Cosmed Quark RMR metabolic cart (Cosmed, Rome, Italy) with a ventilated canopy hood was used to measure oxygen consumption and carbon dioxide production. The calorimeter was serviced and calibrated daily according to the manufacturer's instructions. All REE measurements adhered to evidence-based guidelines for resting metabolic rate assessment (*Compher et al., 2006*), ensuring that subjects were fasted, rested, and had refrained from intense exercise prior to testing (Table 1).

## Grip strength measurements

Upper limb muscular strength was evaluated by measuring handgrip strength (GS) using a digital handgrip dynamometer (Takei T.K.K. 5401, Takei Scientific Instruments, Tokyo,

**Table 1  Evidence-based guidelines for measurement of resting metabolic rate with IC (*Compher et al., 2006*).**

| Criteria | Guidelines for measurement | Study group recommendation |
|---|---|---|
| Fasting (thermic effect of food) | Minimum fast 5 h after meals or snacks (Grade II), 4 h after small meal if longer fast is clinically inappropriate (Grade II) | All recommendations concerning preparations for the study were outlined, including: having rest minimum for 20 min, abstention from nicotine for minimum 2 h, refraining from the consumption of meals 12 h before the test, refraining from drinking beverages with caffeine and alcohol content for the last 48 h before the test, as well as refraining from participation in a physical activity for the previous 14 h. The method of conducting the study was explained in detail and each study participant had the opportunity to visit the test rooms beforehand and familiarize themselves with the equipment so that it did not raise concerns or cause anxiety in the researched group. |
| Alcohol ingestion | Minimum abstention from alcohol for 2 h (Grade III) | |
| Nicotine ingestion | Minimum abstention from nicotine for 2 h (Grade II) | |
| Caffeine ingestion | Minimum abstention from caffeine for 4 h (Grade II) | |
| Rest periods | Rest 10–20 min (Grade III) | |
| Physical activity restriction | Minimum abstention from moderate aerobic or anaerobic exercise for 2 h before test (Grade II), for vigorous resistance exercise abstention of at least 14 h (Grade III) | |
| Environmental conditions | Allow a room temperature of 20 °C–25 °C (68 °F–77 °F) (Grade III) Ensure each individual is physically comfortable with measurement position during the test and repeated measures are in the same reclined position (Grade V) | The rooms had a controlled temperature between 22 to 25 °C. In addition, each participant had the opportunity to acclimatize in the environment by lying flat for 30 min. |
| Gas collection devices | Use rigorous adherence to prevent air leaks (Grade III). Further studies comparing modern gas collection devices are needed in healthy and clinical populations (Grade V) | With these devices, exhaled gas was captured by a canopy (ventilated hood system) or a face mask connected to oxygen and carbon dioxide analyzers mounted on a metabolic cart. This is essential for correct measurement. |
| Steady-state conditions and measurement interval | Discard initial 5 min. Then achieve a 5-minute period with 10% CV[b] for $VO_2$[c] and $VCO_2$[d] (Grade II) | We use a 20-minute protocol in which the first 5 min of data are discarded and the remaining 15 min of data have a coefficient of variation of no more than 10%. |
| No. of measures/24 h | Achieve steady state and one measure is adequate; if not, two to three nonconsecutive measures improve accuracy (Grade II) | 1 measure/24 h |
| Repeated measures (daily to monthly variation) | Repeated measures vary 3%–5% over 24 h (Grade II) and vary up to 10% over weeks to months (Grade II) | – |
| Respiratory quotient (RQ) | RQ measures 0.70 or 1 suggest protocol violations or inaccurate gas measurement (Grade II) | The Quark RMR is a state-of-the-art metabolic system designed for accurate measurement of Resting Energy Expenditure (REE) and respiratory ratio (R), in a non-invasive way, through the measurement of oxygen consumption (VO2) and carbon dioxide production (VCO2) together with other ventilatory parameters. RQ was between 0.7 and 1.0. |

**Notes.**
[a]Grade I, strong, consistent evidence; Grade II, somewhat weaker evidence and disagreement among authors may exist; Grade III, limited design quality; Grade IV, professional opinion only, no clinical trials; Grade V, no available studies.
[b]CV, coefficient of variation (standard deviation [mean of individual replicate measures] × 100).
[c]$VO_2$, oxygen consumption.
[d]$VCO_2$, carbon dioxide production.

Japan). Before testing, the dynamometer's grip span was adjusted for each participant according to the manufacturer's instructions. Each participant performed three maximal grip trials with their dominant hand, squeezing the dynamometer as hard as possible for (tsim3 s per trial *Mehmet, Yang & Robinson, 2020*). A 60-second rest interval was given between trials. The testing procedure followed the standardized protocol from the National Health and Nutrition Examination Survey (NHANES) (*Centers for Disease Control and Prevention (CDC), 2013*), with participants standing upright, feet shoulder-width apart, arm at the side (not touching the body), shoulder neutral, and elbow extended during each attempt.

### Sample size and statistical power

No *a priori* sample size calculation was performed for this observational study. However, a *post hoc* power analysis (G*Power v3.1) indicated that the final sample ($N = 189$) was sufficient. Given the observed between-group differences in REE and lean mass (effect size $d \approx 0.65$), the study achieved a power $> 0.85$ at $\alpha = 0.05$, confirming that the sample size was adequate to detect meaningful differences.

### Ethics approval and informed consent

This study was conducted in accordance with the ethical standards of the Declaration of Helsinki. The research protocol received approval from the Institutional Bioethics Committee of the University of Rzeszów (Resolution No. 2/01/2019). Written informed consent was obtained from all participants and their parents or legal guardians prior to data collection.

### Data analysis

Descriptive statistics (mean, standard deviation, median, and interquartile range) were calculated for all variables. The Shapiro–Wilk test was used to assess normality of data distributions. Because many variables were not normally distributed, between-group comparisons were made using non-parametric tests. Specifically, a Kruskal–Wallis one-way ANOVA was used to compare the four player-position groups, and if a global difference was detected, *post hoc* pairwise comparisons (with appropriate adjustment) identified which groups differed. Categorical variables were compared using the chi-square test. All analyses were performed using Statistica v13.0 software (TIBCO Software Inc., Palo Alto, CA, USA), with a significance level set at $p < 0.05$.

## RESULTS

The study involved 189 boys, aged 9–19 years ($\bar{x} = 14.7 \pm 2.6$ years). The mean body mass was $58.4 \pm 16.1$ kg, and the mean height was $166.9 \pm 14.5$ cm.

The group was divided according to the players' position into four groups: attackers ($N = 43$), defenders ($N = 70$), goalkeepers ($N = 21$), and midfielders ($N = 55$). Details are presented in Table 2.

Significant differences in age, body mass, height, BMI, GS, lean mass (kg), fat mass (kg), BMC, BMD, and REE were found between the positions of the players.

**Table 2  Comparison of selected parameters in children according to position.**

| | Total N = 189 | | | | | Attacker (a) N = 43 | | | Defender (d) N = 70 | | | Goalkeeper (g) N = 21 | | | Midfielder (m) N = 55 | | | p |
|---|---|---|---|---|---|---|---|---|---|---|---|---|---|---|---|---|---|---|
| | $\bar{x}$ | SD | Me | Q25 | Q75 | Me | Q25 | Q75 | Me | Q25 | Q75 | Me | Q25 | Q75 | Me | Q25 | Q75 | |
| Age (years) | 14.7 | 2.6 | 14.6 | 12.4 | 17.0 | **14.0**[m] | 11.6 | 15.5 | 14.7 | 12.4 | 16.8 | 15.6 | 13.5 | 17.3 | **15.2**[a] | 13.1 | 17.4 | **0.0300** |
| Body mass (kg) | 58.4 | 16.1 | 59.1 | 45.0 | 71.3 | **51.3**[g] | 37.1 | 64.4 | **58.7**[g] | 44.3 | 70.1 | **73.3**[a,d] | 61.3 | 81.1 | 61.4 | 48.0 | 72.6 | **0.0011** |
| Height (cm) | 166.9 | 14.5 | 169.0 | 154.5 | 178.0 | **164.0**[g] | 148.0 | 176.0 | 169.5 | 154.5 | 178.0 | **176.0**[a] | 171.0 | 184.0 | 169.0 | 159.0 | 178.0 | **0.0232** |
| BMI | 20.3 | 3.0 | 20.3 | 18.1 | 22.1 | **19.4**[g,m] | 17.1 | 20.4 | **20.2**[g] | 17.9 | 22.5 | **22.9**[a,d] | 21.2 | 24.3 | **20.8**[a] | 18.4 | 23.0 | **0.0001** |
| GS (kg) Left hand | 28.9 | 10.9 | 30.0 | 19.0 | 37.5 | 23.0 | 16.0 | 31.0 | 30.5 | 19.0 | 39.0 | 32.0 | 27.0 | 40.0 | 33.0 | 22.0 | 39.0 | **0.0413** |
| GS (kg) Right hand | 30.3 | 11.2 | 31.0 | 20.0 | 40.0 | **23.0**[m] | 17.0 | 34.0 | 31.5 | 20.0 | 40.0 | 35.0 | 27.0 | 40.0 | **35.0**[a] | 24.0 | 40.0 | **0.0247** |
| Lean mass (kg) | 45.67 | 13.70 | 48.74 | 32.94 | 57.39 | **40.45**[g] | 29.25 | 53.17 | **47.43**[g] | 31.85 | 56.68 | **57.60**[a,d] | 49.78 | 62.91 | 49.23 | 34.72 | 57.02 | **0.0033** |
| Fat mass (kg) | 10.52 | 5.49 | 9.44 | 7.60 | 12.18 | **8.47**[g] | 7.03 | 9.83 | 9.55 | 7.52 | 12.58 | **11.58**[a] | 9.02 | 18.16 | 9.62 | 7.50 | 12.36 | **0.0030** |
| Fat mass (%) | 18.8 | 6.1 | 17.5 | 14.1 | 21.7 | 16.6 | 13.7 | 21.2 | 18.3 | 14.2 | 21.7 | 18.6 | 15.0 | 24.8 | 16.9 | 14.2 | 20.9 | 0.7658 |
| BMC (g) | 2,492.8 | 798.2 | 2,526.6 | 1,742.3 | 3,164.3 | **2,232.1**[g] | 1,478.5 | 2,978.7 | 2,510.5 | 1,696.2 | 3,072.1 | **3,182.0**[a] | 2,458.7 | 3,501.6 | 2,644.4 | 1,966.0 | 3,173.2 | **0.0118** |
| BMD (g/cm²) | 1.1 | 0.2 | 1.1 | 0.9 | 1.2 | **1.0**[g] | 0.8 | 1.2 | 1.1 | 0.9 | 1.3 | **1.2**[a] | 1.0 | 1.3 | 1.1 | 0.9 | 1.2 | **0.0386** |
| REE (kcal) | 1,866.7 | 356.0 | 1,890.0 | 1,593.0 | 2,128.0 | **1,729.0**[g] | 1,471.0 | 1,994.0 | 1,880.0 | 1,548.0 | 2,128.0 | **2,088.0**[a] | 1,833.0 | 2,349.0 | 1,923.0 | 1,715.0 | 2,129.0 | **0.0040** |

Notes.

$\bar{x}$, mean; SD, standard deviation; Me, median; Q25, Q75, quartiles; BMI, body mass index; GS, grip strength; BMC, bone mineral content; BMD, bone mineral density; REE, resting energy expenditure; p, results for Kruskal–Wallis Test. Significant associations are highlighted in bold. *Post hoc* analysis results for multiple comparisons (two-sided), significant associations are highlighted in bold.

The *post hoc* analysis results for multiple comparisons are shown in Table 2.

In terms of age, attackers had the lowest age (14.0 years) and were statistically significantly lower than midfielders (15.2 years). Body mass differed significantly between attackers and goalkeepers (51.3 kg *vs.* 73.3 kg) and defenders and goalkeepers (58.7 kg *vs.* 73.3 kg). Goalkeepers were the ones who had the highest body weight. A statistically significant height difference was observed between attackers and goalkeepers (164 cm *vs.* 176 cm).

The GS was statistically significant only for the right hand and differed between attackers and midfielders (23 kg *vs.* 35 kg).

We found statistically significant differences in body composition for body fat (kg) and lean body mass (kg). Lean body mass differed significantly between attackers and goalkeepers (40.45 kg *vs.* 57.60 kg) and defenders and goalkeepers (47.43 kg *vs.* 57.60 kg). Body fat differed significantly between attackers and goalkeepers (8.47 kg *vs.* 11.58 kg).

We have determined bone mineral density. Age-appropriate $z$-score values for only one participant indicated low BMD ($z$-score $= -1.3$). The other participants did not show abnormalities in bone mineral density. However, we showed statistically significant differences between the groups. For both BMC (g) and BMD (g/cm²), goalkeepers showed statistically significantly higher values than attackers.

Using indirect calorimetry, we evaluated REE. We noticed statistically significant differences between attackers and goalkeepers (1,729 kcal *vs.* 2,088 kcal).

Table 3 shows the differences between the groups regarding BMI and body fat categories. Participants were categorized into underweight, normal body weight, overweight, and obese according to centile grids for BMI and centile grids for body fat, respectively. There were no significant statistical differences based on weight categories according to body fat percentiles. Therefore, on this basis, it can be concluded that the differences in body weight between groups are explained by differences in lean body mass.

**Table 3  Comparison of BMI and body fat percentiles in children according to position.**

| | | Total N = 189 | | Attacker N = 43 | | Defender N = 70 | | Goalkeeper N = 21 | | Midfielder N = 55 | | p |
|---|---|---|---|---|---|---|---|---|---|---|---|---|---|
| | | N | % | N | % | N | % | N | % | N | % | |
| **BMI percentiles** | Underweight | 3 | 1.6 | | 2.3 | 1 | 1.4 | 0 | 0 | 1 | 1.8 | |
| | Normal body weight | 166 | 87.8 | 41 | 95.3 | 62 | 88.6 | 14 | 66.7 | 49 | 89.1 | **0.0394** |
| | Overweight | 19 | 10.1 | 1 | 2.3 | 6 | 8.6 | 7 | 33.3 | 5 | 9.1 | |
| | Obesity | 1 | 0.5 | 0 | 0.0 | 1 | 1.4 | 0 | 0 | 0 | 0.0 | |
| **Body fat percentiles** | Underweight | 2 | 1.1 | 1 | 2.3 | 0 | 0 | 1 | 4.8 | 0 | 0 | |
| | Normal body weight | 140 | 74.1 | 33 | 76.7 | 53 | 75.7 | 12 | 57.1 | 42 | 76.4 | 0.5224 |
| | Overweight | 27 | 14.3 | 5 | 11.6 | 11 | 15.7 | 4 | 19.0 | 7 | 12.7 | |
| | Obesity | 20 | 10.6 | 4 | 9.3 | 6 | 8.6 | 4 | 19.0 | 6 | 10.9 | |

**Notes.**
$p$ –results for chi-square test. Significant associations are highlighted in bold.

On the other hand, we noted statistically significant differences in body weight categories according to BMI percentiles. The highest number of individuals with normal body weight according to BMI percentiles was among attackers (95.3%), and the lowest number was in the goalkeeper group (66.7%).

# DISCUSSION

The purpose of the study was to determine and compare selected variables of body composition, upper limb strength, and resting energy expenditure depending on the position on the football field among young soccer players. To our knowledge, this is one of the few articles that assesses the mentioned relationships in the population of physically active children. Research in this area allows for the identification of criteria for selecting players for specific positions on the field, which is an important aspect of their future sports career.

In soccer players, high tolerance to fatigue, physical performance, aerobic capacity, and the ability to repeat short bursts of high intensity are crucial (*Demo, Senestrari & Ferreyra, 2007*; *Barros et al., 2007*; *Di Salvo et al., 2007*). Furthermore, the position of the player on the field requires specific characteristics and predispositions to fitness, taking into account the somatic structure and overall fitness (*Barros et al., 2007*; *Gore et al., 2008*). Age and somatic parameters are significant criteria considered when identifying the most optimal position for each player on the field and creating individual training plans to optimize sports achievements (*Schuth et al., 2016*). The results were analyzed on the basis of the players' positions on the field. Significant differences in age, body mass, height, BMI, GS, lean mass, fat mass, BMC, BMD, and REE were found between the positions of the players. In soccer, the evaluation of these parameters is particularly useful to determine the desired body structure for a specific position on the field (*Lees et al., 2010*; *Stanula et al., 2009*). According to *Reilly, Bangsbo & Franks (2000)*, anthropometric and physiological criteria play an important role in the selection of players for the appropriate position on the field and in the holistic monitoring of the fitness of young soccer players. In our
study, attackers had the lowest age (14 years) and were statistically significantly different compared to midfielders (15.2 years). The oldest participants were among the goalkeepers (15.6 years). The significance of age-somatic indicators depending on the position in the field is indicated by the results of other studies (*Sebastiá-Rico et al., 2023*). According to *Eider (2003)*, goalkeepers are selected from the oldest, tallest, and best-built players, due to the large size of the goal and the need to meet the challenge of defending the ball during matches.

The results of the *post hoc* analysis revealed significant differences between the body mass of attackers and goalkeepers, as well as defenders and goalkeepers, and a statistically significant difference in height between attackers and goalkeepers. According to *Ważny (2000)*, age, height, and body mass are the three main somatic parameters that influence the outcome of sports performance in soccer. Therefore, goalkeepers or defenders are usually placed with taller, physically stronger players. However, the study of *Koniarek (1969)* suggests that in distant positions from the goal, shorter players with lower body mass often occupy these roles. This is because agility and technical skills are more useful in offensive positions (*Eider, Buryta & Cieszczyk, 2003*; *Gołaszewski & Wieczorek, 2001*). Our study showed statistically significant differences in the body composition of participants in terms of fat tissue, which differed significantly between attackers and goalkeepers, and in terms of lean body mass, which differed significantly between attackers and goalkeepers, as well as defenders and goalkeepers. The results regarding BMI and fat tissue also indicated that the highest proportion of individuals with normal body mass was among attackers, and the lowest was in the goalkeeper group, and these were statistically significant differences.

In football, body composition plays a crucial role in achieving an optimal physical level, which can translate into a good level of play, as the outcomes in this sport depend on various psychophysiological factors. Therefore, depending on the position of the player on the field, their body composition may vary (*Cavia et al., 2019*). Numerous studies on differences in body mass composition based on the player's position on the field have shown that goalkeepers, compared to other positions, tend to have the highest height, body mass, and fat mass (FM) (*Sebastiá-Rico et al., 2023*; *Sutton et al., 2009*). In the study by *Cavia et al. (2019)*, the highest percentage of body fat was observed in goalkeepers, followed by defenders, in relation to other positions (midfielders and attackers). These relationships are confirmed by other studies (*Sebastiá-Rico et al., 2023*; *Ziv & Lidor, 2011*) as well as experts from European football associations (UEFA) (*Collins et al., 2021*). According to a systematic review with meta-analysis conducted by Sebastio-Rico et al., goalkeepers are characterized by the highest height, age, body mass, and muscle mass (MM). Defenders also have the highest age and muscle mass. Meanwhile, midfielders have the lowest height, total mass, and MM. Attackers are the youngest and, similarly to midfielders, have the lowest MM. The researchers did not observe significant differences in BMC, BMD, somatotype, body mass, and percentage of FM (*Sebastiá-Rico et al., 2023*).

The mineral density assessment did not reveal any abnormalities among the subjects, except for one participant (reduced BMD values, $z$-score $= -1.3$). In the case of both BMC and BMD, goalkeepers showed statistically significantly higher values than forwards. Football can be considered a sport that has a positive impact on bone mass during growth.

In the study by *Lozano-Berges et al. (2018)* showed that adolescent football players showed increased bone mass compared to the control group or other athletes. Trainers and specialists can use DXA to monitor the effect of different training schedules on the full body and regional body composition of soccer players at different positions.

The results of indirect calorimetry that evaluated resting energy expenditure (REE) showed statistically significant differences between attackers and goalkeepers (1,729 kcal *vs.* 2,088 kcal). The movement of players during the game and the high variability of their activities require an adequate supply of energy (*Bradley et al., 2009*; *Mohr, Krustrup & Bangsbo, 2003*). This is crucial to maintain the player's optimal body mass, as well as their performance and intensity of actions, at a consistent level throughout the entire duration of the match. As researchers have noted, with the passage of time, the intensity and speed of the actions taken by the players on the field decrease (*Bangsbo, 1994*; *Bradley et al., 2009*; *Mohr, Krustrup & Bangsbo, 2003*). According to *Ali et al. (2007)*, this may be a result of energy deficiencies resulting from improperly calculated energy requirements for these players. Furthermore, proper calorie intake positively affects post-exertion recovery of the body and reduces the risk of injury (*García-Rovés et al., 2014*). According to *Clarke et al. (2011)*, the main causes of fatigue and decreased performance of the players are depletion of glycogen stores in the muscles and liver, as well as dehydration. Therefore, to meet the demands of current football competition, attention must be paid to appropriate energy intake, hydration, and supplementation (*Clarke et al., 2011*; *Anderson et al., 2019*; *Collins et al., 2021*).

While positional differences were observed across several physiological parameters—including resting energy expenditure, lean body mass, and grip strength—we recognize that chronological age may act as a confounding factor. In our cohort, field position assignments often correlate with physical development and performance characteristics, which are, in part, age-related. However, we intentionally designed the study to reflect the real-world structure of youth soccer academies, where age-based heterogeneity and position-specific demands naturally coexist. Thus, the findings should be interpreted in a descriptive and practical context, rather than in pursuit of isolated causal inference.

To account for data characteristics such as non-normal distribution and unequal variances, we adopted a nonparametric statistical framework (Kruskal–Wallis). While more complex models (*e.g.*, ANCOVA or GLM) could be employed to statistically adjust for age, we opted for a conservative analysis to prioritize transparency and clarity. We have provided raw data to enable future researchers to explore more advanced modeling if desired.

In summary, this study has several strengths: it includes a large cohort ($n = 189$); the gold standard of body composition (DXA) was used. This study has several limitations that should be considered when interpreting the findings. First, although all participants were enrolled in standardized training programs at Sports Championship Schools, the age range (9–19 years) introduces biological variability that may influence key physiological and anthropometric measures. While we required a minimum of two years of consistent training and provided a position-based comparison, we acknowledge that age and maturity status were not statistically adjusted in our main analyses. This was a deliberate methodological

decision, as our goal was to reflect position-specific differences observed in real-world settings rather than infer causality.

Second, while nonparametric tests were chosen to handle deviations from normality, no parametric covariate modeling (*e.g.*, ANCOVA) was applied. Future studies might consider using longitudinal or age-matched designs, or implementing statistical controls for age and maturation, to further isolate the effects of playing position.

## CONCLUSIONS

This study provides new insights into the physiological and anthropometric characteristics of youth soccer players across different field positions. By examining body composition, resting energy expenditure, and grip strength in a sample of trained young athletes, we identified position-specific trends that may inform individualized training and development strategies within competitive academy settings.

The use of validated, gold-standard measurement tools—such as DXA and indirect calorimetry—strengthens the reliability of the findings and offers a reproducible protocol for future research. The study's practical relevance lies in its potential to guide coaches, sport scientists, and health professionals in designing age- and role-appropriate conditioning and nutritional programs for young soccer players.

While the cross-sectional design limits causal inference, the results highlight the importance of recognizing both positional and maturational differences in youth athletes. Future longitudinal studies incorporating biological maturity markers and performance metrics could deepen our understanding of how physiological traits evolve with training exposure and positional demands.

### Funding
The authors received no funding for this work.

### Competing Interests
The authors declare there are no competing interests.

### Author Contributions
- Edyta Łuszczki conceived and designed the experiments, performed the experiments, authored or reviewed drafts of the article, and approved the final draft.
- Anna Bartosiewicz conceived and designed the experiments, performed the experiments, authored or reviewed drafts of the article, and approved the final draft.
- Katarzyna Dereń conceived and designed the experiments, performed the experiments, authored or reviewed drafts of the article, and approved the final draft.
- Paweł Jagielski conceived and designed the experiments, analyzed the data, prepared figures and/or tables, and approved the final draft.
- Andrzej Łukasik conceived and designed the experiments, analyzed the data, authored or reviewed drafts of the article, and approved the final draft.

## Human Ethics

The following information was supplied relating to ethical approvals (i.e., approving body and any reference numbers):

The Institutional Bioethics Committee at the University of Rzeszów approved the study (Resolution No. 2/01/2019).

## Data Availability

Raw data is available as a Supplemental File.

## Supplemental Information

Supplemental information for this article can be found online at http://dx.doi.org/10.7717/peerj.19860#supplemental-information.

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
