# Peer review of "Analysis of selected variables in body composition, upper limb strength, and resting energy expenditure among youth soccer players: insights based on field position"

_PeerJ, doi:10.7717/peerj.19860_

## Round 0.1 · original submission · Major Revisions

Dear Authors,

Please revise the manuscript considering reviewer 1 suggestions.

Thank you.

Best regards.

Reviewer 1 ·

Basic reporting

Dear authors, thank you for submitting the interesting manuscript. While I appreciate your purpose, I believe your work needs many improvements. My suggestion follows.

INTRODUCTION (minor revisions): The language is correct, professional, and unambiguous. English was used, and the flow is good. However, too many words have been spent in this section. Introduction may include 3-5 short paragraphs with relevant and recent (if available) findings. Your section sounds too long, and many phrases enforce content already debated. My suggestion is to reduce this section from general to specific content. This part may quickly lead to your results. In addition, relevant recent contributions have been omitted (example https://doi.org/10.3390/biology11060823). Please revise the last decade of research; the references you reported stopped at 2010.

Experimental design

METHODS (Major revisions):
Participants' baseline characteristics sound unrealistic. Did all participants train three times per week in each age category?
What do you mean by "for a minimum of 2 years"? Your participants were 9 to 19, generating a massive biological variability. If players of the same age vary in experience, it sounds difficult to extend your results to other populations.
In line 138 you reported "training 3 times a day/match once a week", but in line 142 you stated "... ~90 min/4 times per week) times per week), training at the gym (exercising motor skills (~45 min/4 times a week) per week) and competing in matches (90 min/once a week)...". Please make your statements coherent. Other investigators may be able to reply to your study in this section. In addition, inclusion and exclusion criteria should be well-stated and clearly stated.

What about the sample size estimation? A priori analysis should be used to select the number of participants needed to reach the desired statistical power.
Line 158: Stature and height are two different measures; please correct it.
Lines 161-169: Please report the reference you used to assess DXA. If you applied your method, please explain it.
STATISTICS (major revisions): In my opinion, authors should assess all the analyses again. A two-way ANOVA with the interaction effect should be performed to take into account age differences. It is not detectable if the difference is due to age or role. In addition, with your sample size, Shapiro-Wilk is expected to easily reject the null hypothesis. If you want to look for residual distributions, other tests are recommended. However, with your sample, it could be omitted (https://doi.org/10.2307/2286841).

Validity of the findings

RESULTS (major revisions): As previously debated, Table 2 enhances where your design failed. It is not possible to detect if your results (differences) are due to role or age. Authors should perform all the statistical analysis and rewrite this section. However, the main text and table reflect the same contents; readers should find in the text info that is not readable in tables and figures. Many efforts are needed.
DISCUSSION and CONCLUSION: Due to the gaps in previous sections, it is not possible to comment before receiving appropriate results.

·

Basic reporting

The manuscript is written in clear, unambiguous, and professional English throughout. The text maintains a high standard of academic language and expression, ensuring clarity and accessibility for an international readership. The authors have structured the article in line with accepted scientific conventions, presenting each section in a logical and coherent manner.

The introduction provides a comprehensive background to the study, appropriately situating the research within the broader context of youth soccer performance and physiological profiling. Relevant and up-to-date literature is well-referenced, offering a solid rationale for the study and clearly identifying the knowledge gap addressed by the research.

Figures and tables are relevant, appropriately labeled, and of sufficient quality to support the findings. They are integrated effectively within the manuscript and contribute meaningfully to the interpretation of results.

The manuscript is self-contained, and all results presented are directly relevant to the stated objectives and hypotheses. The inclusion of comprehensive raw data, as required by the journal’s Data Sharing policy, further enhances the transparency and reproducibility of the work.

Overall, the manuscript meets the journal’s standards for basic reporting and presents a well-organized and clearly communicated body of research.

Experimental design

The manuscript presents original primary research that is well within the aims and scope of the journal. The research question is clearly defined, highly relevant, and addresses a meaningful gap in the literature concerning body composition, strength, and resting energy expenditure in youth soccer players, stratified by field position.

The study is well justified and grounded in a solid understanding of the current state of knowledge. The authors provide a clear rationale for their investigation and explain how their findings contribute new insights that can support both athletic development and targeted training interventions for young soccer players.

The research was conducted to a high technical and ethical standard. The study population is well described, and appropriate inclusion criteria were applied. Ethical approval was obtained from the Institutional Bioethics Committee, and informed consent was secured from all participants and their guardians, which aligns with current ethical norms for studies involving minors.

The methods are described in thorough detail, allowing for replication by other researchers. The use of validated and standardized tools, such as DXA for body composition and indirect calorimetry for REE, supports the technical robustness of the study. The procedures for grip strength assessment and the statistical analysis plan are also clearly explained, further enhancing the methodological transparency.

Overall, the study design is rigorous, ethically sound, and provides a reliable foundation for the interpretation of the results.

Validity of the findings

The findings presented in the manuscript are valid, statistically sound, and well-supported by the data. All relevant raw data have been provided and analyzed using appropriate statistical methods. The analyses are clearly explained, and the results are interpreted with caution and clarity, without overstatement.
Although novelty and impact are not assessed in this section, it is evident that the study offers valuable insights into the physiological and anthropometric profiles of youth soccer players across different field positions—an area that remains relatively underexplored in this specific age group. The rationale for the investigation is well stated, and the replication of certain findings from adult or professional populations in a pediatric athletic context is both meaningful and relevant to the broader literature.
The conclusions are logically derived from the results and are appropriately linked to the original research question. The authors refrain from making unsupported causal claims and instead focus on clearly articulated associations observed in the data. The discussion remains within the bounds of the presented results and highlights the implications in a balanced and responsible manner.
Overall, the study demonstrates strong internal validity, and its findings represent a credible and useful contribution to the field of sports science and pediatric exercise physiology.

Additional comments

This is a well-conducted and clearly presented study that adds valuable data on youth soccer players’ body composition and energy expenditure by field position. The methodology is robust, and the findings are relevant for coaches and sports health professionals.

The manuscript is well-structured, the discussion is thoughtful, and the limitations are acknowledged appropriately. The study offers practical insights and a solid foundation for future research.

---

## Round 0.2 · accepted · Accept

Dear Authors,

I would like to extend my heartfelt thanks for your valuable contribution during the review process and to congratulate you all on the work you've done on this article.

Best wishes for continued excellent work in the future.

Thank you.

Best regards.

Reviewer 1 ·

Basic reporting

Appropriated for publication

Experimental design

Methods have been improved, but statistical analysis is still weak.

Validity of the findings

Conclusions are well stated, despite being based on weak analysis.

Additional comments

While the authors decided not to improve their manuscript with more appropriate statistical analysis, many efforts have been made to meet other requirements. Although I prefer precise and consistent study design and methodologies, I believe the contents reported in the proposed manuscript may be helpful to soccer technicians and investigators.